# Collective cell migration and metastases induced by an epithelial-to-mesenchymal transition in *Drosophila* intestinal tumors

Kyra Campbell[1,8], Fabrizio Rossi[2], Jamie Adams[1], Ioanna Pitsidianaki[1], Francisco M. Barriga [3], Laura Garcia-Gerique[4], Eduard Batlle[2,5], Jordi Casanova [2,6] & Andreu Casali[7,8]

Metastasis underlies the majority of cancer-related deaths yet remains poorly understood due, in part, to the lack of models in vivo. Here we show that expression of the EMT master inducer Snail in primary adult *Drosophila* intestinal tumors leads to the dissemination of tumor cells and formation of macrometastases. Snail drives an EMT in tumor cells, which, although retaining some epithelial markers, subsequently break through the basal lamina of the midgut, undergo a collective migration and seed polyclonal metastases. While metastases re-epithelialize over time, we found that early metastases are remarkably mesenchymal, discarding the requirement for a mesenchymal-to-epithelial transition for early stages of metastatic growth. Our results demonstrate the formation of metastases in adult flies, and identify a key role for partial-EMTs in driving it. This model opens the door to investigate the basic mechanisms underlying metastasis, in a powerful in vivo system suited for rapid genetic and drug screens.

---

[1] Bateson Centre, Department of Biomedical Science, University of Sheffield, Western Bank S10 2TN Sheffield, UK. [2] Institute for Research in Biomedicine (IRB Barcelona), The Barcelona Institute of Science and Technology, Baldiri Reixac, 10, 08028 Barcelona, Spain. [3] Cancer Biology and Genetics Program, Memorial Sloan-Kettering Cancer Center, New York, NY 10065, USA. [4] Developmental Tumor Biology Laboratory, Hospital Sant Joan de Déu, Fundació Sant Joan de Déu, Sant Joan de Déu, Barcelona 08950, Spain. [5] Institució Catalana de Recerca i Estudis Avançats (ICREA), Barcelona 08010, Spain. [6] Institut de Biologia Molecular de Barcelona (IBMB-CSIC), Baldiri Reixac, 10, 08028 Barcelona, Spain. [7] Institut de Recerca Biomèdica de Lleida Fundació Dr. Pifarré (IRBLleida), 25198 Lleida, Spain. [8] These authors contributed equally: Kyra Campbell, Andreu Casali. Correspondence and requests for materials should be addressed to K.C. (email: kyra.campbell@sheffield.ac.uk) or to A.C. (email: acasali@irblleida.cat)

With metastasis being responsible for 90% of cancer-related deaths, there is a pressing need to understand the underlying mechanisms by which tumor cells colonize distant organs[1]. The metastatic cascade is a multistep process, which starts with the local invasion of the primary tumor into adjacent tissue, followed by dissemination of cancer cells and formation of secondary tumors at distant sites[2]. Over the past years, the prevalent view in the cancer field has been that tumor cells exploit the epithelial-to-mesenchymal transition (EMT) to increase their motility and invasive capabilities during the early stages of the metastatic cascade, a process by which epithelial cells acquire mesenchymal characteristics[3–7]. More recently, however, it has been shown that EMT is not necessary for the generation of metastases[8,9], a matter that has raised an intense debate and conflicting views on the importance of EMT in cancer[10–13].

In all, 80–90% of human colorectal cancers (CRCs) are initiated by loss of the tumor-suppressor gene adenomatous polyposis coli (*APC*), which leads to the constitutive activation of WNT signaling pathway. A second common event is activating mutations of KRAS, mutated in 40–50% of human CRCs, which stimulates cell growth by rendering cancer cells independent of epidermal growth factor receptor (EGFR) signaling. Combined mutations in the WNT and EGFR pathways are the most frequent initial events of CRC[14]. There are striking similarities between the guts of mammalians and of *Drosophila*[15] and we and others have previously shown that it is possible to model CRC in *Drosophila* by inducing clonal activation of the Wnt and Ras signaling pathways in the adult midgut[16,17]. These intestinal epithelial Apc-Ras clones were mutant for the negative regulators of the Wnt pathway, *Apc* and *Apc2*, and overexpressed the oncogenic form of Ras, UAS-Ras$^{V12}$. We showed that this genetic manipulation led to the formation of tumor-like overgrowths with many hallmarks of human CRC such as increased proliferation, a block in cell differentiation and cell polarity, and disrupted organ architecture[16]. Notably, we found that these tumors were confined to the gut, never being detected in other tissues[16]. These findings indicate that, as in the case of human CRC, combined *APC* and *Ras* mutations are not sufficient to drive tumor dissemination in *Drosophila*.

Here we leverage this *Drosophila melanogaster* model for CRC to revisit the requirement of EMT in epithelial tumor cells for metastatic dissemination. We find that expression of Snail (Sna) in adult *Drosophila* intestinal Apc-Ras tumors leads to the formation of macrometastases, which shows remarkable parallels to human metastases. We find that Snail activates a partial EMT in tumor cells and that tumor cells undergo collective cell migration and seed polyclonal metastases. While flies and fish have emerged as powerful tools to investigate malignancy and perform large-scale genetic and drug screens, to date studies have been limited by a lack of metastatic models where cells can be followed from primary tumor development to growth of macrometastases in adult organisms. We now overcome this issue as we have developed highly sensitive assays that enable the detection of circulating tumor cells, which, combined with in vivo imaging, makes all steps of the process accessible to analysis.

## Results

**Sna activation triggers formation of macrometastases.** To investigate whether EMT might facilitate the formation of metastases in the Apc-Ras model, we expressed Sna, a master EMT transcription factor[18]. Similarly to Apc-Ras, we found that midgut Apc-Ras-Sna clones increased in size over time, generating tumor-like overgrowths between 2 and 4 weeks after clone induction (Supplementary Fig. 1). Remarkably, and in contrast to Apc-Ras clones, we found Apc-Ras-Sna metastatic tumors

(TMets) in multiple distant locations outside the gut, including the abdomen, thorax, and head (Fig. 1a, b, arrows), as well as the ovaries and legs, within 2–3 weeks of induction (Fig. 1c, arrows). Disseminated tumor cells were also observed to colonize the brain, indicating that they were able to cross the blood–brain barrier (Fig. 1c). Macroscopic TMets, visible externally in whole flies were rare (1.2% of the flies analyzed, Fig. 1d). We conclude that induction of an EMT program in primary tumor cells induces rapid metastatic dissemination in *Drosophila* intestinal tumors driven by *APC* and *Ras* mutations (Fig. 1e).

To test the growth potential of tumor cells present in metastases, we carried out tumor allografts into adult host flies, an assay routinely used to assess the tumorigenic potential of mutant tissues[19]. We dissected metastases present in the abdomen of flies bearing Apc-Ras-Sna clones (Supplementary Fig. 2, arrow) and subsequently transplanted them into adult hosts, where they grew aggressively, and invasively, killing all hosts within 14 days (Fig. 2a, Transplant 1 (T1)). We carried out 10 consecutive rounds of transplantation (T1–T10), during which time the tumors continued to proliferate, implying that metastases have unlimited growth potential. Transplanted tumor cells were invasive, as shown by colonization of the eye (Fig. 2b) or the ovaries (Fig. 2c).

**EMT facilitates dissemination into the hemolymph.** To further characterize the process of primary tumor cell dissemination, we developed a highly sensitive luciferase-based assay to investigate the presence of circulating tumor cells (CTCs). We introduced a UAS-luciferase transgene into the Apc-Ras-Sna genetic background[20] and, as proof-of-principle, analyzed the luciferase activity in lysates from whole flies bearing Apc-Ras-Sna clones. We observed an increase in luciferase activity between 1 and 4 weeks after clone induction (Supplementary Fig. 3a), in accordance with the increase in the number of tumor cells observed in the midgut (Supplementary Fig. 1). To understand how luciferase activity related to cell number, we sorted the GFP$^+$ cells from gut-induced Apc-Ras-Sna clones by fluorescence-activated cell sorting. A linear correlation ($r = 0.9994$, $p = 0.0006$) was observed between the number of cells isolated and the amount of luciferase activity detected, allowing us to detect down to 10 cells (Supplementary Fig. 3b).

We next leveraged this methodology to test for the presence of CTCs in hemolymph extracted from individual flies. First, we measured the luciferase activity in the hemolymph extracted from flies bearing midgut-induced clones of green fluorescent protein (GFP) alone or overexpressing Sna and did not detect any CTCs (Supplementary Fig. 3c). Next, we measured CTCs in Apc-Ras-Sna flies 2, 3, and 4 weeks after clone induction. We observed 6.8% of the flies ($n = 59$) with levels of luciferase indicating >10 CTCs 2 weeks after Apc-Ras-Sna clone induction, a percentage that increased to 15% at 3 weeks ($n = 80$) and 19.4% at 4 weeks ($n = 108$) (Fig. 2d). This shows remarkable parallels with some human cancers; for example, in several human breast cancer series, up to 19% of patients with ductal carcinoma in situ had detectable CTCs. Yet, the risk of developing overt metastases was <1% among these patients[21–25]. We also observed a wide distribution in the number of CTCs per fly, ranging from ten to over a thousand cells per fly (Fig. 2e). Consistently, in sections of whole flies we observed the presence of small TMets that were not visible from outside, ranging from duplets to tens of cells (Fig. 2f). Of note, we also examined the hemolymph of flies bearing clones of just Apc-Ras, and a small number of flies had some cells in the hemolymph, but the percentage of flies bearing CTCs and the amount found was extremely low in comparison to those found in Apc-Ras-Sna flies (Supplementary Fig. 3c). Taken

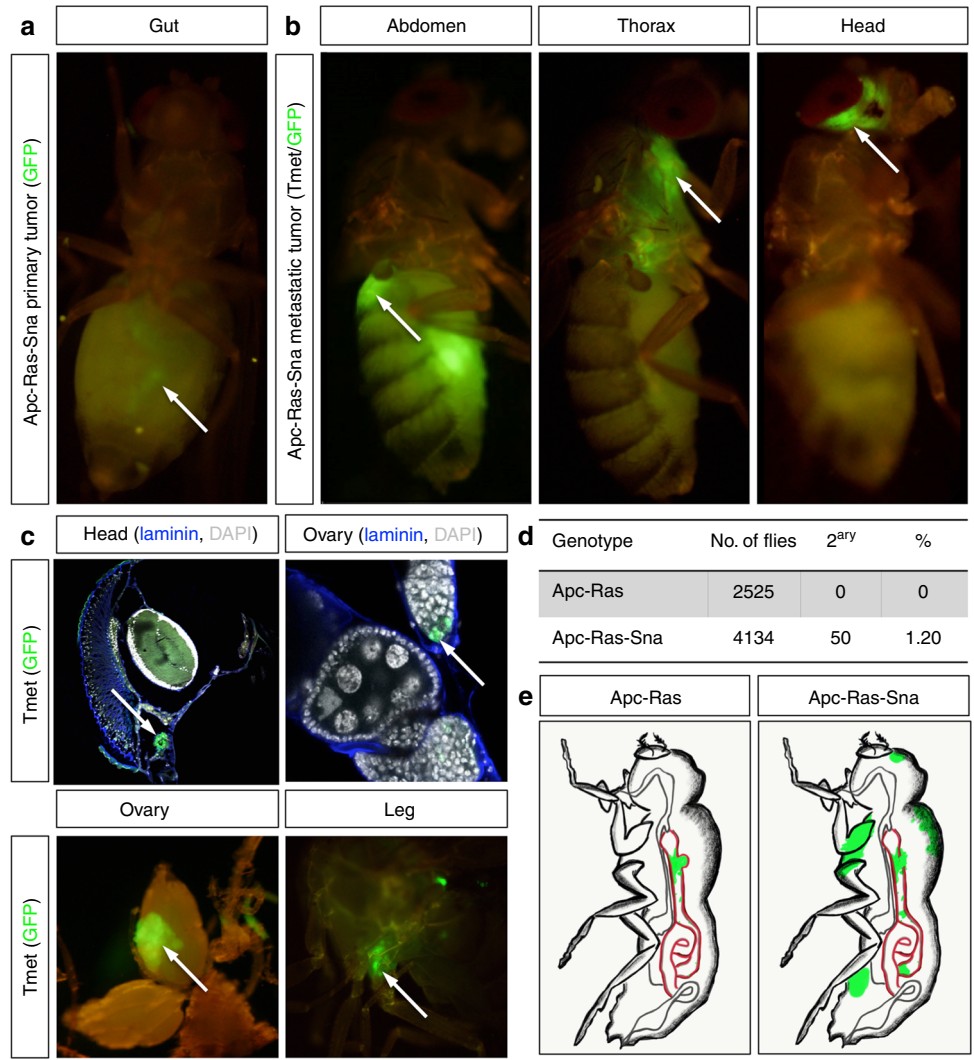

**Fig. 1** Snail expression in Apc-Ras intestinal tumors induces formation of metastases. **a** Apc-Ras-Sna primary tumors, labeled with green fluorescent protein (GFP), are localized in the midgut (arrow). **b**, **c** Apc-Ras-Sna metastatic tumors (TMets) are found in multiple distant locations outside the gut, including the abdomen, thorax, and head (**b**, arrows), and invade other tissues such as the brain, ovaries, and legs (**c**, arrows). **d** Numbers of Apc-Ras and Apc-Ras-Sna flies in which TMets are found. **e** Cartoon depicting Apc-Ras tumors (green) confined to the midgut (red), while Apc-Ras-Sna tumors are able to seed TMets in multiple distant sites

together, these data suggest that Sna overexpression in Apc-Ras clones greatly increases the ability of tumor cells to push out of the gut, invade through the surrounding muscle and basal lamina, and out into the hemolymph.

**Sna induces collective cell migration**. EMT has generally been linked to the dispersion of individual primary tumor cells, yet cancer invasion is often visualized as collective migration of large groups of cells[2,26–29]. We thus investigated how Sna-induced EMT may contribute to the mode of migration and phenotypes adopted by Apc-Ras-Sna cells. We first analyzed the integrity of the basal lamina by immunohistochemical analysis of midguts bearing Apc-Ras and Apc-Ras-Sna clones. As we reported before, Apc-Ras tumors displayed loss of apico-basal polarity, became highly disorganized, and de-localized E-Cadherin[16], but they never appeared to break the basal lamina, as seen by continuous Laminin staining (Fig. 3a). In contrast, high levels of Sna expressed in Apc-Ras cells were associated with large disruption and apparent reorganization of the basal lamina (Fig. 3a, arrow). These images were particularly arresting, as despite the

conceptual importance of basal lamina breakdown for tumor evasion, this phenomenon is rarely visualized in vivo. Concomitantly, we observed groups of GFP$^+$ cells that migrated out of the gut and that were positioned near the areas where the basal lamina has been disrupted, suggesting that its breakdown facilitates collective migration of tumor cells (Fig. 3a, arrow). Furthermore, we occasionally observed primary tumor cells extending out from the gut to envelope surrounding tracheal tubes (Fig. 3b), suggesting that trachea might serve as substrate for their migration, as has been shown in mammalian models of metastasis[30].

Whereas a complete transition from epithelial-to-mesenchymal state is theoretically possible, it has been proposed that tumor cells can undergo a "partial EMT" by which they attain a hybrid epithelial/mesenchymal phenotype. These intermediate states are characterized by a combination of epithelial and mesenchymal features, which may enable collective cell migration (reviewed in refs. [31–33]). Consistent with this notion, we found that Apc-Ras-Sna cells lose epithelial characteristics such as their regular shape and polarity, which can be seen by a dramatic reorganization of F-actin (Fig. 3c) and loss of apico-basal proteins from the cell

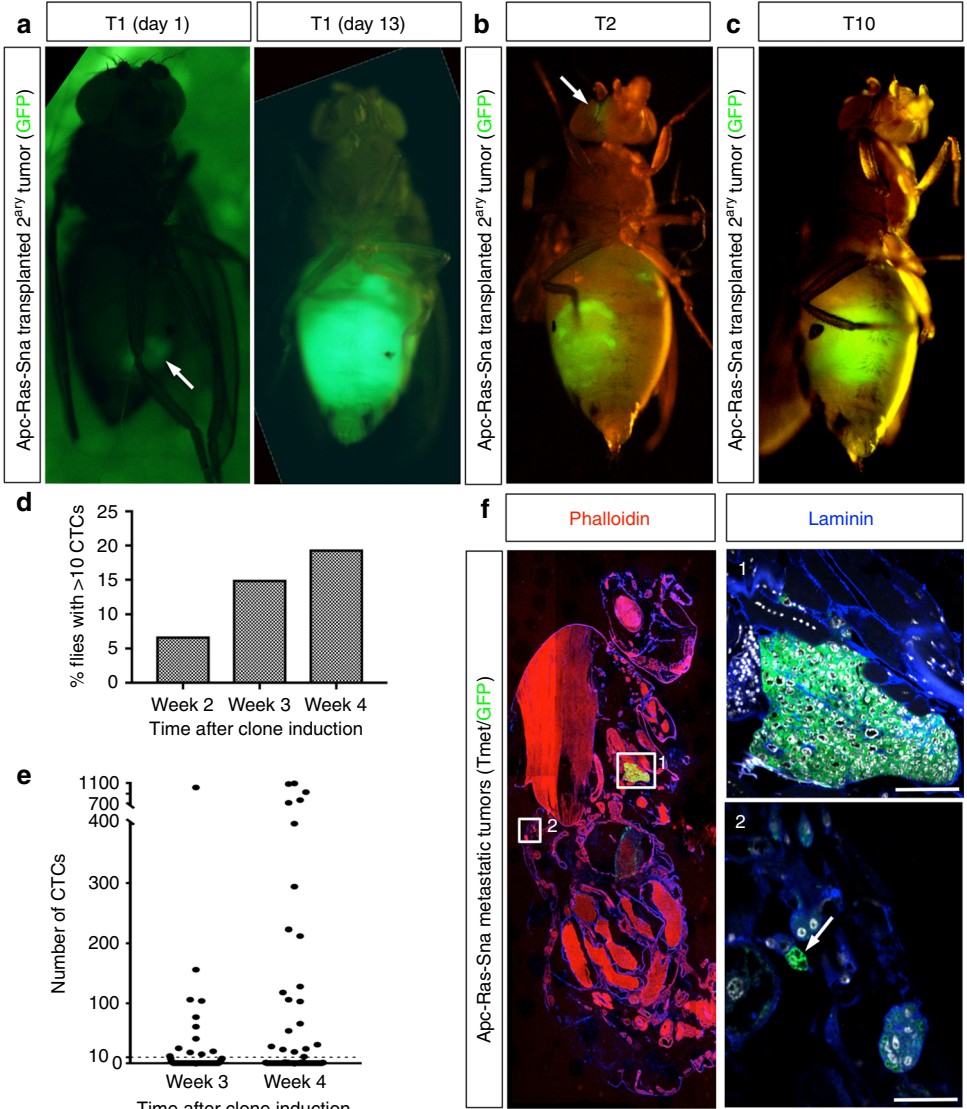

**Fig. 2** Analysis of metastases formation through transplant and circulating tumor cell (CTC) assays. **a** A piece of a green fluorescent protein (GFP)-labeled metastatic tumor (TMet; arrow) implanted into the abdomen of an adult host grows to fill up the entire abdomen of the host fly in just 13 days, rapidly killing the host. **b**, **c** TMets continue to grow during serial implants. In early implants small tumor colonies were seen distant to the abdomen (**b**, arrow), but by T10 TMets became more compact in size and localized in the abdomen (**c**). **d**, **e** Levels of luciferase activity were measured from the hemolymph extracted from individual flies. **d** The percentage of flies found with over 10 CTCs in their hemolymph, $n = 59$ (week 2), $n = 80$ (week 3), $n = 108$ (week 4). **e** The number of CTCs found in the hemolymph of individual flies, measured at 3 and 4 weeks after clone induction. The threshold of cells considered significant (10) is displayed by a dashed line. **f** Sagittal section from an Apc-Ras-Sna fly with a macrometastases in the thorax (1) and small doublets of GFP $^+$ cells near the cuticle of the abdomen (2, arrow). Scale bars: (1) 50 μm and (2) 25 μm

membrane (Fig. 3d, e), and they gain mesenchymal traits, including numerous protrusive membranes (Fig. 4) and the ability to migrate through the basal lamina (Fig. 3a). Yet, at the same time these cells retained the expression of E-Cadherin, which was no longer restricted to the apical membrane but often relocalized around the cell surface or to intracellular punctae (Fig. 3e, arrows). These data are striking when considering a recent study that exploited a lineage-labeled mouse model of pancreatic ducal adenocarcinoma to study for the presence of different EMT imtermediates in vivo. This study revealed that most tumor cells undergo a partial EMT, which they showed is characterized by the internalization and intracellular accumulation of E-Cadherin and other epithelial proteins, rather than transcriptional repression, as well as migration in clusters[34]. Taken together with our observation of groups of tumor cells found outside the gut and associated along the trachea, our results

suggest that Sna overexpression in CRC tumors induces a partial EMT and collective migration of the cells. This type of partial EMT may not be detected by the markers that were recently used to lineage trace EMT processes in mouse models for breast cancer[8], as markers such as fibroblast-specific protein 1 and Vimentin are likely not activated by a cell that only proceeds part-way toward a fully mesenchymal state[11,13].

**Polyclonal composition of metastases.** A prediction from the collective migratory mode observed in Apc-Ras-Sna tumors is that circulating cell clusters seed metastases, and therefore metastases must be polyclonal in origin, as seen recently in mouse models for breast and pancreatic cancer metastases[35–37]. To test this hypothesis, we performed lineage-tracing experiments using the dBrainbow reporter construct[38]. This approach enables the

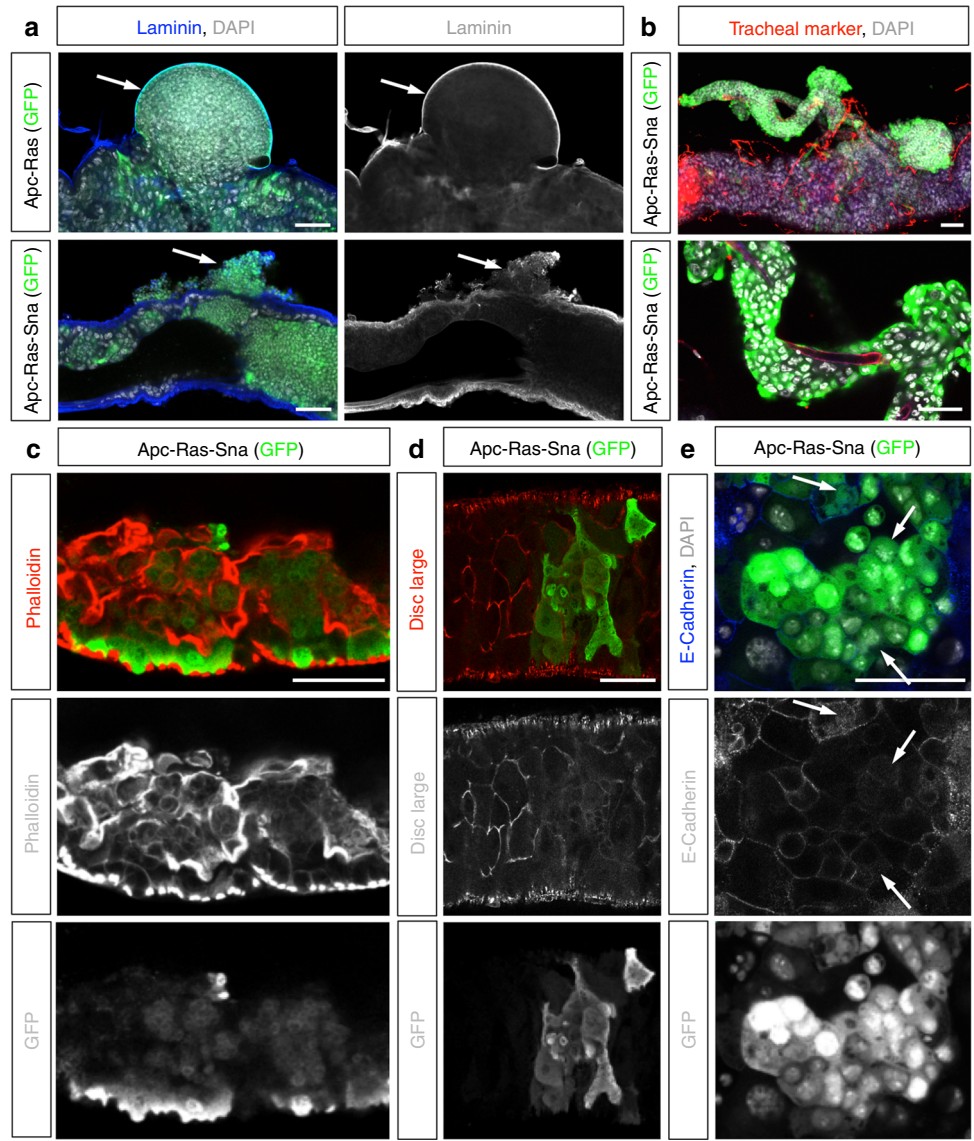

**Fig. 3** Snail induces a partial epithelial-to-mesenchymal transition and collective migration. **a** Apc-Ras (top) and Apc-Ras-Sna (bottom) primary tumors (GFP, Green) and basal lamina (Laminin, blue). APC-Ras primary tumors never break through the basal lamina of the midgut (top, arrow). In midguts bearing Apc-Ras-Sna tumors, the basal lamina is highly disrupted and shows large breaks, with GFP+ cells found outside the midgut near where the basal lamina has been disrupted (bottom, arrow). **b** A primary tumor (GFP, green) extending out from the midgut and enveloping surrounding tracheal tubes (tracheal marker—chitin-binding protein, red). Nuclei are stained with DAPI (white). **c–e** Apc-Ras-Sna primary tumors stained for GFP (green) and F-Actin (**c**, phalloidin, red); discs large (**d**, red) and E-Cadherin (**e**, blue). Arrows in **e** point to intracellular punctae of E-Cadherin. Scale bars = 50 μm

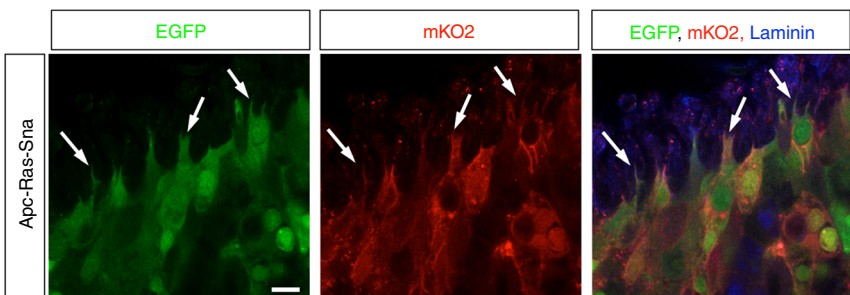

**Fig. 4** Groups of cells in the midgut expressing Apc-Ras-Sna are highly protrusive at their edges. Apc-Ras-Sna tumors labeled with the dBrainbow reporter. Fixed midguts were stained for green fluorescent protein (visualizes enhanced GFP, green), red fluorescent protein (visualizes mKO2, red), and Laminin (blue). Scale bar = 10 μm

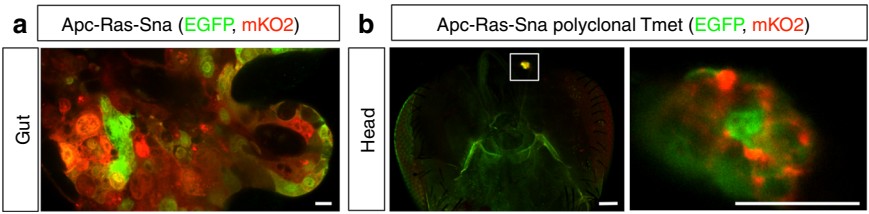

**Fig. 5** Snail induces polyclonal metastases. Apc-Ras-Sna tumors labeled with the dBrainbow reporter. Enhanced green fluorescent protein (green) and mKO2 (red) were imaged in fixed stained midguts (**a**) or unfixed tissues (**b**). Both midgut tumors (**a**) and metastases (**b**) were polyclonal. Scale bars: (**a**, **b** left) 50 µm; (**b** right) 20 µm

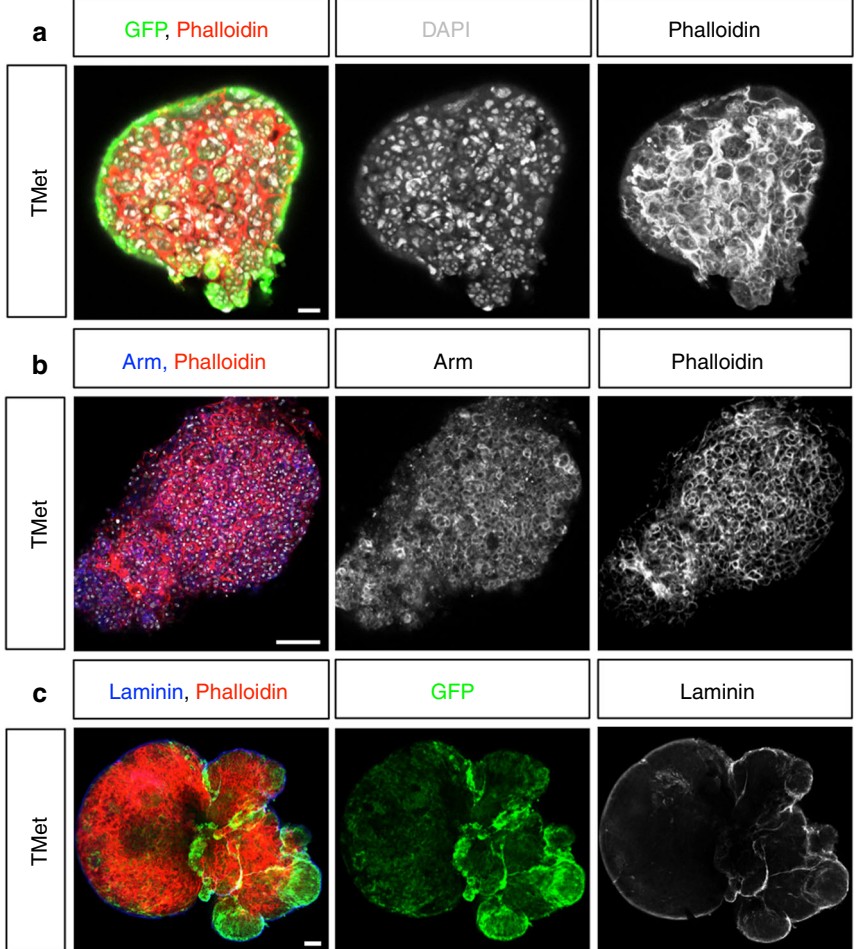

**Fig. 6** Early metastases do not show epithelial traits. Metastatic tumors stained for green fluorescent protein (GFP) and F-Actin (**a**, phalloidin, red); Armadillo (Arm) and F-Actin (**b**, Arm, blue; phalloidin, red); and GFP, Laminin and F-Actin (**c**, Laminin, blue; phalloidin, red). Scale bars: 50 µm

tracing of multi-cell lineages within a single tissue, as the Cre-induced recombination used to generate clones also drives the selection of one of the three fluorescent proteins, with the green and red fluorescence detectable in unfixed samples[38]. We added dBrainbow to the genetic Apc-Ras-Sna background and, following just the green and red fluorescence, found that this gave rise to polyclonal tumors within the midgut (Fig. 5a, Supplementary Fig. 4). We next examined the color make-up of metastases and found that they were multicolored and thus must have had a polyclonal origin (Fig. 5b). This is further supported by the observation that small TMets appear extremely heterogeneous in cell composition, with cells varying greatly in size, shape, and organization (Fig. 6a–c). As previously noted, in a few Apc-Ras flies we observe a low number of CTCs that occur in the absence of any visible breaks in the basal lamina. As we have never

observed TMets in these flies, taken together, our data suggest that large breaks and collective dissemination of tumor cells is required for TMets to form. Thus, while we cannot rule out a role for individual CTCs, these results strongly support a model whereby Sna induces a partial EMT, collective migration of cell clusters, and the seeding of polyclonal tumors. Future live cell imaging studies will help to fully characterize this process of collective cell migration and discern whether individual cell migration and a complete EMT may also contribute to the formation of metastases.

**Re-epithelization of secondary tumors.** Metastases generated by epithelial cancers in humans are often highly epithelial and differentiated. It has been shown that disseminated tumor cells

undergo the reverse process of the mesenchymal-to-epithelial transition (MET) and re-epithelialize to form metastases[25,39–42]. We noted that, upon serial transplants, macroscopic TMets became less aggressive and more compact in shape (Fig. 2b, c), in contrast to transplanted larval brain tumors that drive increased host lethality after repeated transplants[43]. Thus we next aimed to further analyze how they evolved over time. TMets do not display any epithelial characteristics (Fig. 6a–c). Apico-basal polarity markers such as F-Actin and Armadillo are found throughout the cell membrane and the cells appear irregular and highly disorganized (Fig. 6a, b). Staining for basal lamina shows that entire TMets are enclosed in a sheath of lamina, and it is absent from within the tumors (Fig. 6c). We examined E-Cadherin after the TMet had been allowed to grow further and found that, in T1 stages, cells continue to display a non-epithelial organization, with E-Cadherin delocalized throughout cell membranes (Fig. 7a). Upon re-examination after 70 days of growth (in T10), we found that some regions of the tumor exhibited a striking epithelial organization, with E-Cadherin tightly localized to the apical domain of the cells, and lumens clearly visible (Fig. 7a, arrow). Electron microscopic analysis showed the presence of a clear columnar epithelial shape in T10s compared to non-polarized organization of cells in TMets (Fig. 7b). Moreover, T10s contained adherens junctions, apical villi-like structures, lumens, and tracheas within the tumor that were surrounded by basal lamina (Fig. 7c). Consistently, transcriptional analysis confirmed upregulation of epithelial genes in T10 compared to T1 and T2 (Table 1). These results show that, over time, metastases abandon mesenchymal traits and progressively self-organize into complex epithelial structures. The association of tracheal cells with epithelial metastases suggests that they may be recruited to supply oxygen and sustain tumor growth.

These results were very striking as they suggest that MET is not strictly necessary for early stages of metastasis but occurs as the tumors grow. This is surprising given previous studies in mouse models for spontaneous squamous cell carcinomas and breast cancer, which suggested that MET needs to occur in order for metastases to form[25,44]. However, our results are supported by a recent study that identified many intermediate EMT states in skin and mammary primary tumors in mice and found by injecting these isolated cells back subcutaneously into host mice that the cell subpopulations that were best at undergoing MET during tumor metastasis did not correlate with the most metastatic populations[45]. Taken together with our in vivo results, this suggests that other mechanisms than just MET are critical factors for metastatic seeding to occur.

## Discussion

While it has been shown for many years now that model organisms such as flies and fish can develop cancer[46–48], models in which macrometastases develop from primary tumors induced in adult organisms have remained elusive. However, transplant experiments have shown that injected tumor cells do have the capacity to disseminate through adult flies[49,50]. More recently, it was demonstrated that specific genetic mutations in the hindgut leads to the appearance of small foci of tumor cells outside the gut, thus demonstrating that fly tumors can execute early steps of metastasis[51]. Here, by driving constitutive expression of Sna in CRC tumors, we induced cells to break through the muscle and basement membrane surrounding the gut, migrate away, and seed metastases that grew and re-epithelialized over time.

Sna is a master regulator of EMT, which has generally been linked to the dispersion of individual primary tumor cells. However, a number of results suggest that Sna drives the

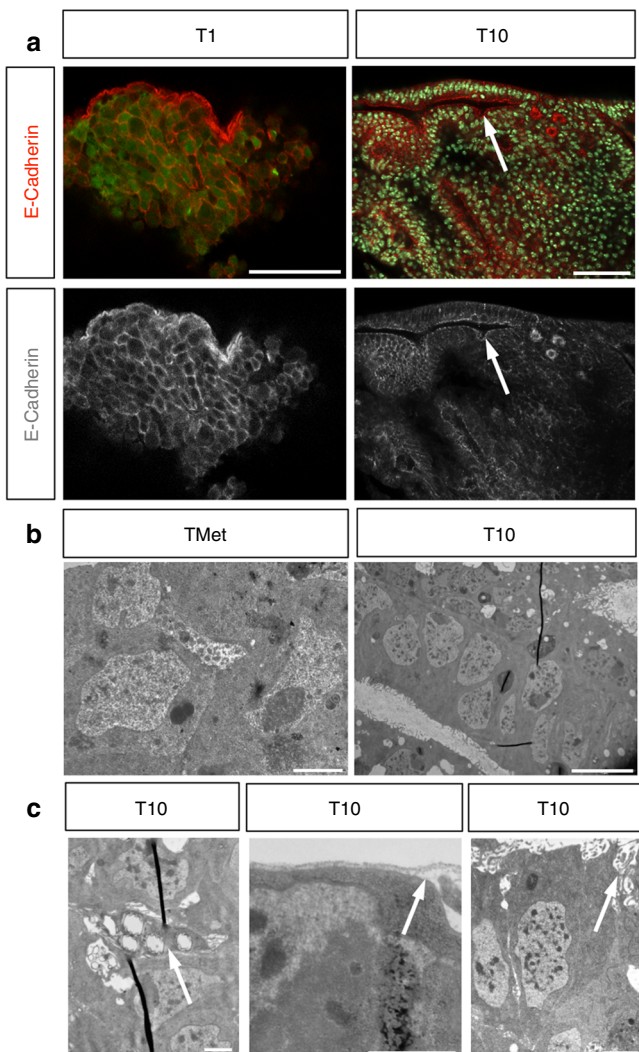

**Fig. 7** Metastases re-epithelialize over time. **a** E-Cadherin localizes throughout the cell membranes at T1 stage (red, left), whereas in T10 it can be found tightly localized to the apical domain of cells, with lumen clearly visible (right, arrow). **b, c** Images taken by electron microscope of a metastatic tumor (TMet) and a stage T10 metastases. **b** Cells in TMets are highly irregular and disorganized (left), whereas in T10 they are organized, with a columnar shape, and form lumen (right). **c** Stage T10 metastases contain trachea within the tumor (left, arrow), are surrounded by a basal lamina (middle, arrow), and make apical villi-like structures (right, arrow). Scale bars: (**a**) 50 µm; (**b**, left) 200 nm; (**b**, right) 5 µm; (**c**) 100 nm

**Table 1 Epithelial genes become highly expressed after several rounds of transplants**

| Gene | Biological function | T1–T10 |
| --- | --- | --- |
| Crumbs | Adherens junctions | 2.16 |
| Bitesize | Apical junction | 4.33 |
| Fat | Cell adhesion | 19.38 |
| Stranded at second | Apical plasma membrane | 25.95 |
| Cadherin 74A | Cell adhesion | 23.27 |
| Vermiform | Chitin binding, trachea | 51.18 |

formation of metastases through activation of a partial EMT and collective cell migration. First, large breaks in the basal lamina near Apc-Ras-Sna clones are associated with groups of tumor

cells, which are found in large clusters outside the gut. Such groups of tumor cells can also often be observed extending out from the midgut and enveloping gut-associated tracheal tubes. Second, groups of tumor cells in the gut are highly protrusive at the edges. Third, lineage-tracing experiments revealed that TMets to be polyclonal, strong evidence that they are seeded by heterogeneous clusters of cells. Finally, the time between the induction of the clones and observation of macrometastases at just 2–3 weeks is extremely short. While all these data strongly suggest that Sna is driving a partial EMT and collective cell migration, it is possible that cells that have undergone a complete EMT and individual cell migration may also contribute to metastases. Future studies using innovative new techniques that enable live imaging of the midguts of adult flies[52] and imaging inside intact adult *Drosophila*[53] will be key to unraveling the relative contributions of each of these cell behaviors.

Even though we cannot recapitulate the entire metastatic cascade[54], as *Drosophila* have an open circulatory system, it is remarkable the number of cellular and organismic aspects of tumorigenesis shared between flies and mammals, despite their divergence over 550 million years ago. Given the power of *Drosophila* genetics and its amenability to state-of-the-art live imaging, this model has enormous potential for dissecting the basic mechanisms underlying tumor dissemination, colonization, and metastatic growth. A unique strength of this model is the ability to visualize the entire process. Combined with assays that act as readouts for each step, rapid screens can be performed, targeted at finding ways to block distinct stages of metastasis through both genetic and pharmaceutical means.

## Methods

**Clone generation**. MARCM clones were generated by a 1-h heat shock at 37 °C of 2–5-day-old females and were marked by the progenitor cell marker escargot (esg) Gal4 line driving the expression of UAS GFP.

**Genotypes**. *yw hsp70-flp; esg Gal4 UAS-GFP UAS-Ras^{V12}/CyO; UAS-Luciferase FRT82B Gal80/TM6b* flies were crossed with *yw hsp70-flp;Sp/CyO; FRT82B Apc2^{N175K}Apc^{Q8}/TM6b* flies to generate Apc-Ras clones and to *yw hsp70-flp;UAS-Sna/CyO; FRT82B Apc2^{N175K}Apc^{Q8}/TM6b* flies to generate Apc-Ras-Sna clones. *yw hsp70-flp; esg Gal4 UAS-GFP/CyO; UAS-Luciferase FRT82B Gal80/TM6b* flies were crossed with *yw hsp70-flp;UAS-Sna/CyO; FRT82B/TM6b* flies to generate Sna clones and to *yw hsp70-flp;Sp/CyO; FRT82B/TM6b* flies to generate GFP clones. Apc2^{N175K} is a loss-of-function allele, Apc^{Q8} is a null allele, UAS-Ras^{V12} is a gain-of-function transgene, and UAS-Sna is a wild-type transgene. Stocks were obtained from Bloomington Stock Center and VDRC. UAS-dBrainbow was a gift from Stefan Luschnig[55].

**Staining and antibodies**. Adult female flies were dissected in phosphate-buffered saline (PBS). All the digestive tract was removed and fixed in PBS and 4% electron microscopic-grade paraformaldehyde (Polysciences, USA) for 35 min. Samples were rinsed 3 times with PBS, 4% bovine serum albumin (BSA), and 0.1% Triton X-100 (PBT-BSA) and incubated with the primary antibody overnight at 4 °C and with the secondary antibody for 2 h at room temperature. Finally, the samples were rinsed 3 times with PBT-BSA and mounted in DAPI-containing media (Vectashield, USA). All the steps were performed without mechanical agitation. Primary antibodies were mouse anti-Armadillo (1:100, Hybridoma Bank, N2 7A1); mouse anti-Discs large (1:500; Hybridoma Bank, 4F3); rat anti-E-Cadherin (1:100, Hybridoma Bank, DCAD2); goat anti-GFP (1:500; Abcam, ab6673); rabbit anti-Laminin (1:500, Abcam, ab47651); mouse anti-RFP (1:300; Life Technologies, MA5–15257). Secondary antibodies were from Invitrogen (USA). TRICT-conjugated Trachael chitin was visualized with CBP (chitin-binding protein) at 1:300 (a gift from Jordi Casanova). Phalloidin (Sigma, USA, P1951) was used at 5 μg/ml. Confocal image were acquired with a Leica SP5 or Zeiss LSM 880. Images were analyzed with the Fiji software [National Institutes of Health (NIH) Bethesda, MD] and assembled into figures using Fiji, the Adobe Photoshop software, and Microsoft Powerpoint.

**Luciferase assays**. Luciferase assays were performed using the Dual-Luciferase(R) Reporter Assay System. Samples were loaded into 96-well plates and read on a Varioskan plate reader. For whole-fly lysates, flies were squashed using a pipette tip into the Luciferase buffer. For hemolymph analysis, hemolymph was extracted

from whole flies according to the instructional video published by Laura Musselman (https://www.youtube.com/watch?v=im78OIBKlPA).

**Transplants**. Three-week-old adult flies bearing Apc-Ras-Sna clones showing macroscopic TMets visible under the scope were washed in distilled water and dissected in 0.7% NaCl on a siliconized microscope slide and cut into five small pieces. Young female adult hosts (genotype was yw) were anesthetized with $CO_2$ and stuck on a microscope slide, ventral side up, with double-sided sticky tape. Each piece of a TMet was picked up with the tip of a glass needle and injected tangentially in the mid-ventral part of the abdomen of one adult host. Implanted hosts were kept at 29 °C. For the next round of transplantations, a transplanted fly was dissected, the GFP+ tumor mass was cut into several small pieces and injected again in five adult host flies. This process was repeated for 10 rounds.

**Gene expression analysis**. TMets (GFP+) were dissected from Apc-Ras-Sns flies between 3 and 4 weeks after induction and incubated for 15 min at 65 °C in Lysis buffer (20 mM DTT, 10 mM Tris.HCl ph 7.4, 0.5% sodium dodecyl sulfate, 0.5 μg/μl proteinase K). RNA was isolated with a RNA clean XP Kit (Agencourt Bioscience). cDNA synthesis and amplification was performed by a TransPlex® Complete Whole Transcriptome Amplification Kit (WTA2–50RXN, Sigma-Aldrich).

**Microarray processing**. Microarray samples from each experiment were processed separately using packages affy[56] and affyPLM[57] from R[58] and Bioconductor[59]. Raw cel files were normalized using RMA background correction and summarization[60]. Technical metrics described by ref. [61] were computed and recorded as additional features for each sample. Standard quality controls were performed in order to identify abnormal samples and relevant sources of technical variability[62] regarding: (a) spatial artifacts in the hybridization process (scan images and pseudo-images from probe-level models); (b) intensity dependences of differences between chips (MvA plots); (c) RNA quality (RNA digest plot); (d) global intensity levels (boxplot of perfect match log-intensity distributions before and after normalization and RLE plots); and (e) anomalous intensity profile compared to the rest of samples (NUSE plots, Principal Component Analyses). No samples were excluded according to the results of these quality-control checks. Chip probeset were annotated using the information provided by Affymetrix.

**Reporting summary**. Further information on research design is available in the Nature Research Reporting Summary linked to this article.

## Data availability

The microarray data have been deposited in the Gene Expression Omnibus (GEO) database under the accession code GSE125312. All the other data supporting the findings of this study are available within the article and its supplementary information files and from the corresponding authors upon reasonable request.

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

## Acknowledgements

We are thankful to the IRB Cell separation Unit and the Electron cryo-microscopy laboratory from the Scientific & Technological Centre (University of Barcelona) and to Bloomington stock center for fly stocks. We also thank the Casanova, Llimargas, Araujo, Franch-Marro, Evans, Brown, Zeidler, Strutt, and Bulgakova laboratories for discussions; Y. Rivera for technical help; and Lucy O'Brian, Robert Weinberg, Diwakar Pattabiraman, Jackson Liang, Orla Conneely, and Katarina Hadjantonakis for comments on the manuscript. This study was supported by grants from the Ministerio de Economía y competitividad (Grant Numbers BFU2014–59781-P (to A.C.), BFU2015–73494-JIN (to K.C.), BFU2015–66488-P (to J.C.)). This work was also supported by the Joseph Steiner Foundation (to E.B.) and a Wellcome Trust/Royal Society Sir Henry Dale Award to K.C. (Grant number R/148777–11–1).

## Author contributions

K.C. and A.C. initiated this study, designed the experiments, performed most of the experiments and analyses, and wrote the paper. F.R., J.A., I.P., F.M.B., and L.G.-G. helped with experiments. F.R. performed allograft assays and provided samples for microarray analyses. J.A. carried out analyses of gut tumor burden and also statistical analyses and interpretation of data. E.B. and J.C participated in the data interpretation. All authors reviewed the manuscript.

## Additional information

**Competing interests:** The authors declare no competing interests.

