## [Peer Review File · Nature Communications]

Reviewers' comments:

Reviewer #1 (Remarks to the Author):

In this manuscript, the authors showed that Snail overexpression drives a partial-EMT, which is required for the break of basal lamina of the midgut in a *Drosophila* tumor model. The authors further found that these tumor cells undergo a collective migration and seed polyclonal metastases, and that MET event is not strictly required for the localization and metastatic growth. Overall, data are clean, conclusive and well presented. Although some the ideas are not novel and have been shown in other modeling systems, the findings in this manuscript provide solid evidence to illustrate the critical function of Snail and EMT in metastasis in vivo, which is under heated debate in the field. Therefore, this reviewer supports the publication of this study.

Reviewer #2 (Remarks to the Author):

Campbell's manuscript demonstrated the first model for tumor metastasis in adult flies and showed that Snail expression in APC/Ras tumor intestine cells can lead to cancer invasion and metastasis to other tissues and organs. In addition, they showed that Snail expression induces partial EMT and collective cell migration for metastases, early metastatic tumors are more mesenchymal while after several transplantations, tumors will become more epithelial. This is an interesting and important manuscript for providing another potentially useful in vivo model to study tumor invasion and metastasis. Unfortunately, their main conclusion on partial EMT and collective cell migration for metastases, and also many other important results have been demonstrated by their "descriptive" figures, which lacks both strict and enough controls and also necessary statistical quantification. Thus, it is difficult to conclude whether the Snail-induced tumor metastases are really from collective cell migration/invasion, or are indirectly from individually invading cells which might disrupt and break basal laminin limitation to facilitate the proliferation and growth of tumor groups out of original intestines. It is possible that after individual tumor cells invade and move to new tissues or organs, they might unite together and proliferate to mix with each other, so tumors in secondary sites look like a collective cell behavior.

Main concerns:

1. In Figure 1, authors showed and compared the metastatic ability of Snail/APC/Ras tumor cells vs. APC/Ras tumor cells. How about only Snail expressing normal intestine cells? Can they do metastases? Since Snail is an important gene for metastasis, while it can't support tumor growth.

2. In Figure 2d, authors quantified the calculation of flies with more than 10 CTCs. Although author showed some quantification data in extended data figure 3, it is unclear how the equation between luciferase activity and cell number is achieved? It seems that in extended data figure 3b, authors only accessed 3 data (one possibly near 10 cells, one near 100 cells, and the other near 500 cells). The absence of other cell numbers might lead to the unprecise equation. In addition, it is unclear which is the minimal cell number that this luciferase assay can inform? More detailed information can tell us how precise this quantification in Figure 2d can get. Another important issue is that if this method can detect flies with around 10 CTCs, it means that in most cases, CTCs in each flies could be mostly individual, considering that CTCs might be diluted into circulation system in each flies. If this is true, data of CTC number could strongly contradict the main conclusion of collective cell migration authors argued.

3. Consistent with CTC number, it seems that individually invading cells can be easily detected in many figures, such as green cells in Figure 2F right panels (laminin figures 1 and 2 top regions), Figure 3d right-top corner, Figure 3b trachea, Figure 3e (2-3 individual cells on top). Although authors showed that tumor cell groups in secondary sites are heterogeneous, it is still possible that individual cells invade to new places and then to proliferate and fuse with each other. The presence of individual Snail/APC/Ras cells might indicate that these tumor cells might do individual cell migration and invasion, and after breaking basal laminin limitation, they might unite together or fuse with the overproliferating tumor cell groups. Thus, without precise control experiments or more dynamic or informative figures, it is difficult to conclude that collective cell migration mediates metastases. To clarify this, authors should express LifeActRFP in these green-colored Snail/APC/Ras cells to see whether lamellipodia structures are present in leading or free edge of tumor cell groups. They can also check whether they can find a precise time window (such as a few hours) when tumor cells start to invade basal laminin, so that they can show and quantify which types of cells are really migrating and invading basal laminin, by fixed imaging or possibly live imaging.

4. In Figure 3a, authors showed that Snail/APC/Ras tumor cells are together when basal laminin is damaged. However, it is unclear whether this is due to individual cell invasion firstly, tumor cells grow out secondly, or due to collective cell migration. Authors should try to do chemical treatment after dissection, to see whether after laminin limitation is off, APC/Ras tumor cell groups might have the same or different morphologies as Snail/APC/Ras tumor cells.

5. It is very unclear how authors concluded a partial EMT induced by Snail. In Figure 3c-e, it is unclear whether and which type partial EMT occurs. From Figure 3e, 3 types of E-cadherin phenotype seem to be present: E-cadherin adhesion is still present but not in apical, E-cadherin is situated to cytosol possibly in intracellular vesicle, or E-cadherin level is strongly reduced. Authors just mentioned that E-cadherin redistribution occurs in these cells, thus indicating a partial EMT process. It is unclear whether this redistribution means cytosol or other plasma membrane but not apical. Since authors mentioned in the introduction that APC/Ras tumor cells have a block in polarity, does this mean that APC/Ras tumor cells already lost apical-basolateral polarity? However, author didn't show any figures about APC/Ras tumor cells, stained by F-actin staining, Dlg and E-cadherin. Thus, it is unknown whether E-cadherin adhesion might already redistribute into other apical-basolateral polarity domain. From 3 phenotypes of E-cadherin, we can't exclude the possibility of complete EMT, or EMT with different ranges might be present.

6. In Figure 4, APC/Ras tumor cells as a critical control are completely missing. Without these control experiments, it is unclear whether apical-basolateral polarity is already lost in APC/Ras tumor cells.

Minor concerns:

1. In extended data figure 1, fluorescence in week2 tumor cells are even stronger than that in week3, how to explain this?
2. In Figure 2a-c, authors described the decrease of metastasis ability of Snail/APC/Ras tumors cells after transplantation. It is better to do statistical quantification too.
3. Some figures are difficult to tell green-colored tumor cells from neighboring WT cells, such as Figure 3c. Authors should show individual green channel in this case.
4. If possible, authors should quantify the occurrence chance of metastasis tissue/organs shown in Figure 1c-e, rather than a cartoon to simple summary.

Reviewer #3 (Remarks to the Author):

In this manuscript Campbell et al., present and characterise a *Drosophila* model of Colorectal cancer (CRC) metastasis driven by loss of *Apc*, and overexpression of activated Ras and the EMT transcription factor Snail.

This manuscript presents a very innovative model to study CRC metastasis in flies. This has been a great limitation in the fly gut field. As such, this represent a great contribution to the research field with great potential to impact research in CRC using flies and beyond. The authors present a wide range of novel techniques to characterise the system, such as the assessment of circulating tumours cells, which will be very useful for future directions into this type of research, such as the identification of modifiers of CRC metastasis through genetics and drug screens.

I suggest some experiments that I believe will improve the manuscript and provide stronger support for the main conclusions drawn by this work.

- 1- The addition of some important controls is necessary for some key experiments. For example:
 - a- The data in Figure S1 would greatly benefit from a side by side comparison with Apc, Ras only and Snail only overexpressing clones.
 - b- Similarly, the experiments in Figure 2 need to include controls of transplantations of at least Apc, Ras only tumours. If Snail over expression clones show any phenotype on their own, they should be included in this experiment as well.
 - c- Equally, appropriate controls are needed for the assessment of circulating tumour cells. This would also help to validate the technique and discard any possible presence of gut derived cells in the haemolymph due to technical procedures rather than actual metastasis.
- 2- More quantifications are needed such as the ones presented in Figure 1d. For example for Figures 2c and 3b.
- 3- Are there any possible mesenchymal markers expressed in metastatic cells? Staining or transcript expression of mesenchymal markers in addition to partial loss of E-Cadherin would be helpful to better characterise these tumours as being half-way through EMT.
- 4- Additional characterization of the brainbow system in the gut (Figure 3f) would be useful. Showing earlier gut tumours before the merging of clones due to overgrowth would help visualise individual clones derived from different stem cells.
- 5- The stainings shown in Figure 4d would benefit from separation of channels to better show the description of the E-cad staining patterns provided in the text.

Manuscript: NCOMMS-18-17074

"Collective cell migration and metastases induced by an epithelial-to-mesenchymal transition in *Drosophila* intestinal tumors"

Point by point responses to the reviewers

Reviewer #1 (Remarks to the Author):

In this manuscript, the authors showed that Snail overexpression drives a partial-EMT, which is required for the break of basal lamina of the midgut in a *Drosophila* tumor model. The authors further found that these tumor cells undergo a collective migration and seed polyclonal metastases, and that MET event is not strictly required for the localization and metastatic growth. Overall, data are clean, conclusive and well presented. Although some the ideas are not novel and have been shown in other modeling systems, the findings in this manuscript provide solid evidence to illustrate the critical function of Snail and EMT in metastasis in vivo, which is under heated debate in the field. Therefore, this reviewer supports the publication of this study.

We thank the reviewer for the support to the publication of our work.

Reviewer #2 (Remarks to the Author):

Campbell's manuscript demonstrated the first model for tumor metastasis in adult flies and showed that Snail expression in APC/Ras tumor intestine cells can lead to cancer invasion and metastasis to other tissues and organs. In addition, they showed that Snail expression induces partial EMT and collective cell migration for metastases, early metastatic tumors are more mesenchymal while after several transplantations, tumors will become more epithelial. This is an interesting and important manuscript for providing another potentially useful in vivo model to study tumor invasion and metastasis. Unfortunately, their main conclusion on partial EMT and collective cell migration for metastases, and also many other important results have been demonstrated by their "descriptive" figures, which lacks both strict and enough controls and also necessary statistical quantification. Thus, it is difficult to conclude whether the Snail-induced tumor metastases are really from collective cell migration/invasion, or are indirectly from individually invading cells which might disrupt and break basal laminin limitation to facilitate the proliferation and growth of tumor groups out of original intestines. It is possible that after individual tumor cells invade and move to new tissues or organs, they might unite together and proliferate to mix with each other, so tumors in secondary sites look like a collective cell behavior.

Main concerns:

1. In Figure 1, authors showed and compared the metastatic ability of Snail/APC/Ras tumor cells vs. APC/Ras tumor cells. How about only Snail expressing normal intestine cells? Can they do metastases? Since Snail is an important gene for metastasis, while it can't support tumor growth.

This is an interesting point also raised by referee number 3. We have now carried out this experiment, generating flies that bear clones over-expressing Snail alone, or GFP alone. We have found that expression of neither Snail nor GFP alone is sufficient to induce the ability of intestinal cells to metastasize out of the gut, as these cells are not found in the haemolymph. We have included this data in Extended Data Fig. 3c.

2. In Figure 2d, authors quantified the calculation of flies with more than 10 CTCs. Although author showed some quantification data in extended data figure 3, it is unclear how the equation between luciferase activity and cell number is achieved? It seems that in extended data figure 3b, authors only accessed 3 data (one possibly near 10 cells, one near 100 cells, and the other near 500 cells). The absence of other cell numbers might lead to the unprecise equation. In addition, it is unclear which is

the minimal cell number that this luciferase assay can inform? More detailed information can tell us how precise this quantification in Figure 2d can get.

In order to determine the relationship between luciferase activity and number of cells, we sorted GFP⁺ cells from guts bearing Apc-Ras-Sna clones by FACS. We isolated 3 batches of exactly 10 cells, 3 batches of 100 cells and 3 batches of 500 cells and measured their luciferase activity. We averaged the results for the three replicas and the resulting graph, as seen in Extended Data Fig. 3b, showed a linear correlation of 0.9994 and p-value of 0.0006 between the number of cells isolated and the amount of luciferase activity detected. The striking linear correlation of near 1 and low p-value suggests that these data points were sufficient to provide a clear linear graph which can be used to precisely correlate luciferase activity with cell number, at least between 10 and 500 cells.

In any case, the reviewer is indeed right that it may be possible to use the luciferase assay to detect fewer than 10 cells. Our data show that we can reliably detect 10 cells, as we have counted them by FACS and tested for luciferase activity. However, it would be difficult to differentiate a lower number of cells from background noise, and therefore we decided to take 10 cells as the minimal amount of cells to consider that a fly is positive for CTCs. Indeed, the current degree of sensitivity has allowed us to test for the presence of CTCs in the haemolymph of individual flies in different conditions. We find that the haemolymph extracted from flies with intestinal clones expressing GFP alone, or Snail alone, never gives a value above the 10-cell threshold. In contrast, in flies bearing Apc-Ras-Sna clones we can detect anything from 10 to over a thousand cells. These results strongly suggest that Sna greatly increases the ability of tumor cells to push out of the gut, invade through the surrounding muscle and basal lamina, and out into the haemolymph.

Another important issue is that if this method can detect flies with around 10 CTCs, it means that in most cases, CTCs in each flies could be mostly individual, considering that CTCs might be diluted into circulation system in each flies. If this is true, data of CTC number could strongly contradict the main conclusion of collective cell migration authors argued.

We agree with the referee that these assays do not allow distinguishing between single cells and groups of cells. However, in analysing these data some observations have to be considered. On the one hand, in most cases (80% at 4 weeks) Apc-Ras-Sna flies do not show detectable levels of CTCs in their haemolymph. On the other hand, when we do detect CTCs, the number in any individual fly ranges from over 10 to over a thousand cells. Thus we think this data in itself does not contradict the idea that collective migration is driving metastasis formation. On the contrary, taken together with the following additional observations, we think strongly supports our conclusion that collective cell migration contributes to metastases formation:

i) The most striking difference between Apc-Ras and Apc-Ras-Sna midguts was the finding of large breaks in the basal lamina near Apc-Ras-Sna clones, something that was never seen in Apc-Ras midguts. These breaks were always associated with groups of tumour cells that appear to be moving collectively outside the gut, a finding that was reinforced by the observation of groups of cells extending out from the midgut and enveloping tracheal tubes.

ii) Lineage tracing experiments using the dBrainbow construct led to the discovery of polyclonal TMets, strongly suggesting that they may be seeded by heterogeneous clusters of cells.

iii) Following the suggestion of the reviewer in comment 3, we examined groups of cells in the gut for protrusions, and found them highly protrusive at the edges of groups of cells in the gut, again pointing to a collective nature to the migration of these cells.

However, we would like to point out that while our data strongly supports that collective migration contributes to metastasis formation, none of these pieces of evidence completely rule out a role for individual cell migration in contributing also to the seeding of TMets, as we comment in the paper.

3. Consistent with CTC number, it seems that individually invading cells can be easily detected in many figures, such as green cells in Figure 2F right panels (laminin figures 1 and 2 top regions), Figure 3d right-top corner, Figure 3b trachea, Figure 3e (2-3 individual cells on top). Although authors showed that tumor cell groups in secondary sites are heterogeneous, it is still possible that individual cells invade to new places and then to proliferate and fuse with each other. The presence of individual Snail/APC/Ras cells might indicate that these tumor cells might do the individual cell migration and invasion, and after breaking basal laminin limitation, they might unite together or fuse with the overproliferating tumor cell groups. Thus, without precise control experiments or more dynamic or informative figures, it is difficult to conclude that collective cell migration mediates metastases. To clarify this, authors should express LifeActRFP in these green-colored Snail/APC/Ras cells to see whether lamellipodia structures are present in leading or free edge of tumor cell groups. They can also check whether they can find a precise time window (such as a few hours) when tumor cells start to invade basal laminin, so that they can show and quantify which types of cells are really migrating and invading basal laminin, by fixed imaging or possibly live imaging.

While we see individual cells in the gut, as the reviewer pointed out in the top-right Fig. 3d, and in Fig. 3e, we have almost never detected these outside the gut. For example, in Fig. 2f2, the cells with the arrow pointing to it are a small group of cells, and with the other cells the reviewer pointed to, it is difficult to say if these are really individual or not. Also, the presence of individual cells does not support nor contradict the role of collective cell migration in the seeding of metastasis. As mentioned above, our data strongly supports that collective migration contributes to metastasis formation, does not rule out that individual cell migration may also contribute. We agree with the referee that live imaging would greatly help to understand how cells are migrating outside the primary tumor. However, live imaging of adult midguts has remained one of the biggest challenges in the field. An innovative new approach has been described in a recently published paper (Martin JL, Sanders EN, Moreno-Roman P, Jaramillo Koyama LA, Balachandra S, Du X, O'Brien LE (2018). Long-term live imaging of the *Drosophila* adult midgut reveals real-time dynamics of division, differentiation, and loss. *Elife*. 2018 doi: 10.7554/eLife.36248). In an attempt to address the issues raised by the reviewer we have tried this technique and found that while it shows promise, it relies on having clones invade a very specific section of the midgut – R4. After attempting to image in a number of flies, we mapped the location of Apc-Ras-Sna clones in several flies and found that they do not tend to arise in this specific R4 region (see below for an example). We hope in the future to be able to further develop this technique and be able to analyse sections R2 and R5, but as it is a great challenge we feel it is outside the scope of this paper.

We have also followed the suggestion of the referee to look for lamellipodia structures at the leading or free edge of tumour cell groups. We examined groups of cells in the gut for protrusions, and found

them highly protrusive at the edges. We include images of this new experiment in Extended Data Fig. 4.

4. In Figure 3a, authors showed that Snail/APC/Ras tumor cells are together when basal laminin is damaged. However, it is unclear whether this is due to individual cell invasion firstly, tumor cells grow out secondly, or due to collective cell migration. Authors should try to do chemical treatment after dissection, to see whether after laminin limitation is off, APC/Ras tumor cell groups might have the same or different morphologies as Snail/APC/Ras tumor cells.

We agree with the referee that particular experiments, when taken individually, might be explained by alternative interpretations, which in some cases we think are much unlikely. However, we think that seeing tumour cells together out of the basal lamina once it is damaged strongly supports them coming out together. We are afraid that the experiment suggested here by the referee, technically very challenging, would not solve the issue as, again, different alternative interpretations could be raised regarding the role of the chemical treatment to the observed similarity or differences in cell morphologies. We note here that the conclusion of our experiments is to point out to the role of collective migration in Sna-triggered metastasis rather than completely discarding a contribution of individual migrating cells. In this regard, all experiments considered, we think our data clearly support a role for collective migration.

5. It is very unclear how authors concluded a partial EMT induced by Snail. In Figure 3c-e, it is unclear whether and which type partial EMT occurs. From Figure 3e, 3 types of E-cadherin phenotype seem to be present: E-cadherin adhesion is still present but not in apical, E-cadherin is situated to cytosol possibly in intracellular vesicle, or E-cadherin level is strongly reduced. Authors just mentioned that E-cadherin redistribution occurs in these cells, thus indicating a partial EMT process. It is unclear whether this redistribution means cytosol or other plasma membrane but not apical. Since authors mentioned in the introduction that APC/Ras tumor cells have a block in polarity, does this mean that APC/Ras tumor cells already lost apical-basolateral polarity? However, authors didn't show any figures about APC/Ras tumor cells, stained by F-actin staining, Dlg and E-cadherin. Thus, it is unknown whether E-cadherin adhesion might already redistribute into other apical-basolateral polarity domain. From 3 phenotypes of E-cadherin, we can't exclude the possibility of complete EMT, or EMT with different ranges might be present.

This point of how to better characterise this as a partial-EMT versus a complete-EMT was also raised by referee number 3. We have decided to use the partial-EMT category as we have observed co-existence of epithelial and mesenchymal features in these cells. On the one hand, these cells express epithelial markers. However, we could not assess the expression of the classic mesenchymal markers in other systems such as Vimentin, Fibronectin and FSP1 as these are not expressed in *Drosophila*. For this reason and as in previous studies (eg. Campbell et al 2011), we have characterised the mesenchymal state by a transition to a mesenchymal morphology, and a gain in motility and protrusive activity – all of which we see in the Apc-Ras-Sna expressing tumour cells. We have added a figure showing numerous protrusions at the edges of groups of cells in the gut, and include images of this in the new Extended Data Fig. 4. We also comment on the dramatic reorganization of F-Actin (Fig. 3c) and the ability of Apc-Ras-Sna cells to degrade and migrate through the basal lamina (Fig. 3a). In contrast, as mentioned in the text, while Apc-Ras tumours do indeed have a block in polarity (this was documented in Martorell 2014, we have added this reference where we comment on polarity) we never observed breaks in basal lamina, nor migration outside the gut, indicating that the cells do not activate these mesenchymal traits.

Moreover, while it has increasingly been recognised that EMT encompasses a range of hybrid phenotypes called partial-EMTs, what a partial-EMT actually is has not been well defined in molecular terms. Since submitting this paper, a study was published which exploited a lineage-labeled mouse model of pancreatic ductal adenocarcinoma to study for the presence of different EMT-

intermediates *in vivo*. This study revealed that most tumor cells undergo a partial-EMT, which they showed is characterized by the internalization and intracellular accumulation of E-Cadherin and other epithelial proteins, rather than transcriptional repression, as well as migration in clusters. Remarkably, in Apc-Ras-Sna primary tumors we find that cells retained expression of E-Cadherin, which was not restricted to the apical membrane, but often relocalised around the cell surface or to intracellular punctae. We have added a discussion and reference to this new paper to the text, as well as arrows to intracellular punctae of E-Cadherin in Fig. 3e.

Taken together we feel that this strongly endorse our use of partial-EMT to characterize these tumours.

6. In Figure 4, APC/Ras tumor cells as a critical control are completely missing. Without these control experiments, it is unclear whether apical-basolateral polarity is already lost in APC/Ras tumor cells.

In Figure 4 we are showing TMets from Apc-Ras-Sna clones. Apc-Ras TMets would clearly be good to compare for a control but we could not rely on them, as Apc-Ras clones do not form TMets. Of course we could compare TMets from Apc-Ras-Sna clones with Apc-Ras primary tumours in the gut, but these are so different that would not be useful as controls.

Minor concerns:

1. In extended data figure 1, fluorescence in week2 tumor cells are even stronger than that in week3, how to explain this?

There is a high variability in the tumor burden shown by flies bearing Apc-Ras-Sna clones at each time point. To clarify this point we have quantified extensively the number of GFP⁺ cells across 10 guts in each condition at each week, and have included this data in a completely new Extended Data Fig. 1. Of note, we have also added controls for GFP clones alone, Sna clones alone and Apc-Ras clones at each time point.

2. In Figure 2a-c, authors described the decrease of metastasis ability of Snail/APC/Ras tumors cells after transplantation. It is better to do statistical quantification too.

Yes, the author is right, but we would like to emphasise that the transplantation experiments were performed in order to show that TMets have a proliferative capacity beyond the normal life span of a fly. In doing the experiments, we also observed that transplanted tumors seemed to be more compacted after several transplantation rounds, but this was a qualitative observation that is not amenable to statistical quantification. However, we thought worth pointing it out.

3. Some figures are difficult to tell green-colored tumor cells from neighboring WT cells, such as Figure 3c. Authors should show individual green channel in this case.

We have now added an individual green channel for figures 3c,d and e in Extended Data Fig. 4.

4. If possible, authors should quantify the occurrence chance of metastasis tissue/organs shown in Figure 1c-e, rather than a cartoon to simple summary.

When screening for TMets, our main goal was to chart where TMets could arise. As a future study we aim to chart the probability of TMets arising in different locations in the fly and if it does not appear to be random, to understand the underlying mechanisms, but we feel that this is outside the scope of this paper.

Reviewer #3 (Remarks to the Author):

In this manuscript Campbell et al., present and characterise a *Drosophila* model of Colorectal cancer (CRC) metastasis driven by loss of Apc, and overexpression of activated Ras and the EMT transcription factor Snail. This manuscript presents a very innovative model to study CRC metastasis in flies. This has been a great limitation in the fly gut field. As such, this represents a great contribution to the research field with great potential to impact research in CRC using flies and beyond. The authors present a wide range of novel techniques to characterise the system, such as the assessment of circulating tumour cells, which will be very useful for future directions into this type of research, such as the identification of modifiers of CRC metastasis through genetics and drug screens. I suggest some experiments that I believe will improve the manuscript and provide stronger support for the main conclusions drawn by this work.

1-The addition of some important controls is necessary for some key experiments. For example:

a- The data in Figure S1 would greatly benefit from a side by side comparison with Apc, Ras only and Snail only overexpressing clones.

In response to comments from both referee 2 and 3 we have included more controls – please see our new Extended Data Fig. 1 showing the tumour burden, clone number and distribution of clones in guts bearing GFP alone, Sna alone, Apc-Ras and Apc-Ras-Sna clones at 1,2,3 and 4 weeks. As there is a big variability in clone size and number shown by flies bearing clones in each of these genotypes, we have also quantified extensively the number of GFP⁺ cells (as % of the whole gut that is covered with GFP) across 10 guts in each condition at each week.

b- Similarly, the experiments in Figure 2 need to include controls of transplantations of at least Apc, Ras only tumours. If Snail over expression clones show any phenotype on their own, they should be included in this experiment as well.

Transplantations in Fig. 2 were of secondary metastases, which are not found in Apc-Ras flies. For a control from Apc-Ras flies primary tumours in the gut would have to be used. We and others have tried to transplant primary CRC tumours into host abdomens, as we have done for secondary Mets, but we find that this kills the host flies – likely due to the presence of bacteria in the transplanted gut. Similarly, we do not find secondary metastases in Sna-overexpressing clones alone, and the gut clones cannot be transplanted for the same reason.

c- Equally, appropriate controls are needed for the assessment of circulating tumour cells. This would also help to validate the technique and discard any possible presence of gut derived cells in the haemolymph due to technical procedures rather than actual metastasis.

We have carried these out and now include these data in Extended Data Fig 3. It includes CTC numbers for GFP alone, Sna alone, Apc-Ras and Apc-Ras-Sna for individual flies at 2 and 3 weeks.

2- More quantifications are needed such as the ones presented in Figure 1d. For example for Figures 2c and 3b.

For transplanted secondary metastases (figure 2c), 100% of the successfully transplanted pieces of secondary metastases grew to take up the entire abdomen. We have added this to the text.

Regarding other quantifications, we would like to point out that in Fig 1d we quantified the number of flies showing macroscopic TMets that can be detected *in vivo*. Therefore, thousand of flies could be screened over a long period of time. The kind of events as the one shown in Fig 3b are quite difficult to quantify as they require the dissection of the fly, and therefore conclusions cannot be taken as we cannot determine whether such an event could have happen if the fly had live longer. To avoid this kind of problems we decided to use the CTCs as an objective way to measure and quantify the number of flies that show metastatic events.

3- Are there any possible mesenchymal markers expressed in metastatic cells? Staining or transcript expression of mesenchymal markers in addition to partial loss of E-Cadherin would be helpful to better characterise these tumours as being half-way through > EMT.

This point of how to better characterise this as a partial-EMT versus a complete-EMT was also raised by referee number 2 and we reproduce here the same response given above. Co-expression of epithelial and mesenchymal markers is a signature of a partial-EMT. However, there is a lack of mesenchymal markers in *Drosophila*, as classic markers in other systems such as Vimentin, Fibronectin and FSP1 are not expressed in *Drosophila*. For this reason in previous studies (eg. Campbell et al 2011), we have characterised the mesenchymal state by a transition to a mesenchymal morphology, and a gain in motility and protrusive activity – all of which we see in the Apc-Ras-Sna expressing tumour cells. We have added a figure showing numerous protrusions at the edges of groups of cells in the gut, and include images of this in the new Extended Data Fig. 4. We also comment on the dramatic reorganization of F-Actin (Fig. 3c) and the ability of Apc-Ras-Sna cells to degrade and migrate through the basal lamina (Fig. 3a). In contrast, as mentioned in the text, while Apc-Ras tumours do indeed have a block in polarity (this was documented in Martorell 2014, we have added this reference where we comment on polarity) we never observed breaks in basal lamina, nor migration outside the gut, indicating that the cells do not activate these mesenchymal traits.

Moreover, while it has increasingly been recognised that EMT encompasses a range of hybrid phenotypes called partial-EMTs, what a partial-EMT actually is has not been well defined in molecular terms. Since submitting this paper, a study was published which exploited a lineage-labeled mouse model of pancreatic ducal adenocarcinoma to study for the presence of different EMT-intermediates *in vivo*. This study revealed that most tumor cells undergo a partial-EMT, which they showed is characterized by the internalization and intracellular accumulation of E-Cadherin and other epithelial proteins, rather than transcriptional repression, as well as migration in clusters. Remarkably, in Apc-Ras-Sna primary tumors we find that cells retained expression of E-Cadherin, which was not restricted to the apical membrane, but often relocalised around the cell surface or to intracellular punctae. We have added a discussion and reference to this new paper to the text, as well as arrows to intracellular punctae of E-Cadherin in Fig. 3e.

Taken together we feel that this strongly suggests that these tumours undergo a partial-EMT rather than a complete-EMT.

4- Additional characterization of the brainbow system in the gut (Figure 3f) would be useful. Showing earlier gut tumours before the merging of clones due to overgrow would help visualise individual clones derived from different stem cells.

We have added a better characterization of Apc-Ras-Sna midguts labelled with the dBrainbow reporter, showing midguts at 1,2,3 and 4 weeks in Extended Data Fig 6, as well as an enlargement of the anteriormost set of clones in each midgut shown.

5- The stainings shown in Figure 4d would benefit from separation of channels to better show the description of the E-cad staining patterns provided in the text.

This has been added to the Figure.

Reviewers' comments:

Reviewer #2 (Remarks to the Author):

Thank the authors for addressing many important points. These include the explanation of luciferase assay and quantification, the reasons why some experiments cannot be quantified or feasible to do. Important controls such as Snai or GFP-expressing flies are also included.

Several points need to be somehow improved, mainly in the description of the phenotypes:

1, although all these strongly improve the revised manuscript, collective cell migration is still a weakness of manuscript, which is mainly due to the missing or not-strong data in supporting the real definition of collective cell migration. The global observation could be the integration of many different behaviors, such as individual cell invasive migration (possibly complete EMT), many tumor cell proliferation and division during cancer cell invasion (which might overcome cell apoptosis during metastasis), and of course some collective cell migrations (including some partial-EMT). Thus, we strongly suggest that authors might include all these behaviors in the description of phenotypes observed in Apc-Ras-Snai tumor cells when they do metastasis. It will be also nice to do more discussions (in the discussion section) about how some future studies (such as live cell imaging and other potential experiments, when feasible in technique) will be considered to clarify the detailed behavior and role of collective cell migration in this Apc-Ras-Snai mediated metastasis.

2, about EMT phenotype, the shown images also give us with an impression that it might include a quite big range of EMT phenotypes, from complete EMT to very weak EMT. Authors should not avoid some individual cell invasive behaviors due to complete EMT. If individually migrating cells are fused together later at the secondary sites, the role of complete EMT and individual cell invasion should not be ignored.

3, it is better to also include the images (EGFP, mKO and Laminin) of two important controls, APC-Ras-expressing and only Snai-expressing cells, in the extended data figure 5. Both EGFP and mKO, even not the good marker for F-actin structure in protrusion, seem to be good enough for the detection of protrusive membrane. Here, it is interesting and important to know whether APC-Ras tumor cells have no prominent protrusions (if yes, it will explain why these tumor cells only do proliferation but not do invasion and metastasis), and whether Snai cells have protrusion or not in contact with ECM. It is unclear why Snai only-expressing intestine cells can't do invasion or metastasis, possibly due to the non-proliferation or the apoptosis or other possibilities; and thus characterization of Snai only-expressing cells in invasion will be important. Taken together, this feasible experiment can support the main conclusion of APC-Ras-Snai mediated metastasis in *Drosophila* in vivo, and also help us to better understand the reason of this metastasis. In addition,

only based on EGFP and mKO (not good marker for F-actin structure in protrusions), it is difficult to judge whether filopodia or lamellipodia, or both are present in these protrusive membrane region. Thus, it is more reasonable for the authors to change their sentence to “including numerous protrusive membranes” in page 6.

Reviewer #3 (Remarks to the Author):

This revised manuscript contains experiments addressing most of my concerns and/or additional explanation. I support its publication.

Manuscript: NCOMMS-18-17074

"Collective cell migration and metastases induced by an epithelial-to-mesenchymal transition in *Drosophila* intestinal tumors"

Point by point responses to the reviewers

Reviewer #2 (Remarks to the Author):

Thank the authors for addressing many important points. These include the explanation of luciferase assay and quantification, the reasons why some experiments cannot be quantified or feasible to do. Important controls such as Snai or GFP-expressing flies are also included. Several points need to be somehow improved, mainly in the description of the phenotypes:

1, although all these strongly improve the revised manuscript, collective cell migration is still a weakness of manuscript, which is mainly due to the missing or not-strong data in supporting the real definition of collective cell migration. The global observation could be the integration of many different behaviors, such as individual cell invasive migration (possibly complete EMT), many tumor cell proliferation and division during cancer cell invasion (which might overcome cell apoptosis during metastasis), and of course some collective cell migrations (including some partial-EMT). Thus, we strongly suggest that authors might include all these behaviors in the description of phenotypes observed in Apc-Ras-Snai tumor cells when they do metastasis. It will be also nice to do more discussions (in the discussion section) about how some future studies (such as live cell imaging and other potential experiments, when feasible in technique) will be considered to clarify the detailed behavior and role of collective cell migration in this Apc-Ras-Snai mediated metastasis.

As the reviewer will appreciate, there have been heated debates in the field of EMT about definitions of EMT and of collective migration, and which are the 'real' definitions is contentious. It is for this reason that throughout this work we have tried to accurately describe what we actually observed, and to discuss what the data suggests is most likely to be happening.

In line with this, we agree that besides some partial-EMT and collective cell migrations, there might be other behaviours such as individual cell invasive migration through a complete EMT, or proliferation during cancer cell invasion, which may add to the final phenotype. However, we cannot describe a role for complete-EMT in individual cells as we have not observed it, neither we can describe a role for individual cells contributing to metastasis, as we have not demonstrated it.

On the contrary, we have many lines of evidence that strongly suggest that dissemination is driven by collective migration, including the observation of groups of cells migrating out of the gut, the presence of polyclonal metastases, as well as the short time between the induction of the clones and the observation of the macrometastases. Nevertheless, throughout the text,

we have been careful to discuss when we cannot rule out other scenarios, in particular individual cell migration and complete EMTs.

Due to the short format nature of the paper it is difficult to add extensively to the discussion section. However to address this point, we have added the following two sentences:

'Future live cell imaging studies will help to fully characterize this process of collective cell migration and discern whether individual cell migration and a complete EMT may also contribute to the formation of metastases.'(page 8) and 'Given the power of Drosophila genetics and its amenability to state-of-the-art live imaging, this model has enormous potential for dissecting the basic mechanisms underlying tumour dissemination, cell migration, colonization and metastatic growth.' (page 9)

2, about EMT phenotype, the shown images also give us with an impression that it might include a quite big range of EMT phenotypes, from complete EMT to very weak EMT. Authors should not avoid some individual cell invasive behaviors due to complete EMT. If individually migrating cells are fused together later at the secondary sites, the role of complete EMT and individual cell invasion should not be ignored.

As mentioned above, we do not rule out a role for complete-EMT and individual cell migration contributing to the formation of metastases, and this is something that we have been careful to clearly state in the manuscript. Moreover, we do point out that we have observed small clusters of 2-cells outside the gut (Figure 2f2), suggesting that individual or very small clusters of cells may be able to disseminate from primary tumors.

3, it is better to also include the images (EGFP, mKO and Laminin) of two important controls, APC-Ras-expressing and only Snai-expressing cells, in the extended data figure 5. Both EGFP and mKO, even not the good marker for F-actin structure in protrusion, seem to be good enough for the detection of protrusive membrane. Here, it is interesting and important to know whether APC-Ras tumor cells have no prominent protrusions (if yes, it will explain why these tumor cells only do proliferation but not do invasion and metastasis), and whether Snai cells have protrusion or not in contact with ECM. It is unclear why Snai only-expressing intestine cells can't do invasion or metastasis, possibly due to the non-proliferation or the apoptosis or other possibilities; and thus characterization of Snai only-expressing cells in invasion will be important. Taken together, this feasible experiment can support the main conclusion of APC-Ras-Snai mediated metastasis in Drosophila in vivo, and also help us to better understand the reason of this metastasis. In addition, only based on EGFP and mKO (not good marker for F-actin structure in protrusions), it is difficult to judge whether filopodia or lamellipodia, or both are present in these protrusive membrane region. Thus, it is more reasonable for the authors to change their sentence to "including numerous protrusive membranes" in page 6.

In the last round of revision, in response to comments from both referees 2 and 3 we included a number of new controls, including the tumour burden, clone number and distribution of

clones in guts bearing GFP alone, Sna alone, Apc-Ras and Apc-Ras-Sna clones at 1,2,3 and 4 weeks. We also isolated haemolymph from each of these genotypes and characterised the CTC numbers for GFP alone, Sna alone, Apc-Ras and Apc-Ras-Sna for individual flies at 2 and 3 weeks. All of this data supported the original conclusions in the paper, demonstrating that Apc-Ras-Sna drives the formation of secondary metastases in *Drosophila* in vivo. We do not believe that the results from the suggested experiments would change the conclusions of the paper, or add to our understanding of the mechanism. In particular, we note that the suggested work is addressed towards further characterize Apc-Ras clones (which we already did in our previous manuscript – see Martorell et al 2014, and have referenced this where appropriate) or Sna clones (it is outside the scope of the paper to characterize why Sna clones alone are not able to form metastases). Furthermore, generating suitable fly stocks would take months. It is for these reasons that we argue that these experiments are outside the scope of this work.

We do agree that is difficult to judge between filopodia and lamellipodia and therefore have changed the sentence to protrusive membranes as suggested.

Reviewer #3 (Remarks to the Author):

This revised manuscript contains experiments addressing most of my concerns and/or additional explanation. I support its publication.

We thank the reviewer for the support to the publication of our work.